# Universal Learning of Distribution Flows using Ensemble Control Systems

## Abstract

Constructing interpretable representation models for learning flows of probability distributions has become a thriving research area in machine learning. This effort not only sparks new research avenues but also provides distinctive insights into established fields such as image processing. For example, the recently developed flow matching (FM) model has demonstrated its effectiveness in addressing image inverse problems. However, a unified and comprehensive framework for learning and generating probability flows from data in a general setting remains underexplored. In this work, inspired by an in-depth exploration of FM from a dynamical systems perspective, we develop an ensemble control system (ECS) model for probability flow learning. Our model is represented as an ensemble of heterogeneous control systems, with control inputs acting as time-dependent trainable parameters. The heterogeneous dynamics and time-dependent parameters significantly enhance the model capability of ECS, making it exceptionally powerful. To further capitalize on these strengths, we introduce a moment kernel transform, which generates a reduced kernel representation of ECS over a reproducing kernel Hilbert space, enabling efficient training. We demonstrate the significant advantages of the ECS model through various image restoration tasks and provide a detailed comparison with baseline FM-based image processing models.

## 1    Introduction

The construction of representation learning models for generating desired flows of probability distributions forms the foundation of flow-based generative models (Ho et al., 2019; Bond-Taylor et al., 2022; Tomczak, 2024). The emergence of flow-based models has opened new research avenues in generative artificial intelligence and provided novel insights and tools for well-established fields, notably image processing and generation (Dhariwal & Nichol, 2021; Saharia et al., 2022; Rombach et al., 2022; Podell et al., 2024; Lipman et al., 2023; Ho et al., 2019). The recently developed flow matching (FM) falls into this category of generative models (Lipman et al., 2023; 2024). Integrating with state-of-the-art algorithms for liner inverse problems, pre-trained FM has demonstrated its remarkable capability in various image restoration tasks, including denoising, deblurring, inpainting, and super-resolution(Pokle et al., 2024; Ben-Hamu et al., 2024; Zhang et al., 2024; Martin et al., 2025; Ohayon et al., 2025).

The flow-based characteristic of FM particularly highlights its capability to learn and generate probability flows in a continuous-time manner, which in turn elucidates the intermediacy between FM and dynamical systems. Specifically, from the perspective of dynamical systems theory, the learning objective of FM is the vector field of an ordinary differential equation system whose phase space distribution, that is, the solution of the associated Liouville equation (continuity equation), is the desired probability flow (Lipman et al., 2023; Mei et al., 2025; Brockett, 2007; 2012).

**Our contributions.**    Inspired by the dynamical systems interpretation of FM, we propose a control system representation model for learning and generating flows of probability distributions. Specifically, our model comprises an ensemble of control systems with distinct dynamic characteristics, referred to as an *ensemble control system (ECS)*, in contrast to a single unforced dynamical system used in FM models. By using the control input for targeted manipulation of system dynamics, it naturally stands as the trainable parameter of the proposed ECS model. The time-dependence of

the control input also distinguishes ECS from most machine learning models, where trainable parameters are constant. This feature, combined with the heterogeneity in ECS dynamics, not only promises enhanced model capability but also renders ECS a high-dimensional model. In fact, both the system state space and training parameter space are infinite-dimensional. To mitigate this challenge, we develop a moment kernel transform, which generates a kernel representation of ECS over a reproducing kernel Hilbert space. This ensures efficient training of the ECS model while maintaining satisfactory learning performance. We apply ECS to various image restoration tasks to demonstrate its efficiency and performance, which gives rise to a new dynamic image processing paradigm. Additional, we conducted a detailed comparison with state-of-the-art FM-based image processing models. The main contributions of our work are summarized as follows:

- Construction of an ECS representing model for learning and generating probability flows.

- Development of the moment kernelization framework to enable efficient training of the ECS model without compromising its model capability.

- Application of ECS to image restoration tasks to establish a new dynamic image processing paradigm.

**Related works.** The fundamental objective of this work is the targeted transportation of probability distributions using ensemble control systems. The problem of transporting probability distributions was initially formulated by G. Monge in 1781 (Monge, 1781), which lays the foundation for optimal transport theory (Villani, 2003; 2009; Ambrosio et al., 2008). The connection between optimal transport and dynamical systems was discovered by J.-D. Benamou and Y. Brenier, where probability distributions are transported along flows of dynamical systems (Benamou & Brenier, 1999). Recently, the significance of this dynamical systems-enabled optimal transport was recognized in the realm of generative artificial intelligence, catalyzing the development of FM (Lipman et al., 2023; 2024; Tomczak, 2024).

This emerging FM model particularly provides new insights and tools for image processing. The core idea in this thread of research centers around integrating pre-trained FM with advanced linear inverse algorithms to enhance image restoration performance. State-of-the-art FM-based image processing models include OT-ODE, D-Flow, Flow Priors, and PnP-Flow (Pokle et al., 2024; Ben-Hamu et al., 2024; Zhang et al., 2024; Martin et al., 2025). In OT-ODE, a gradient-based adaptation term is added to the pretrained unconditional flow model, which converts an unconditional vector field to a conditional one to improve the perceptual quality of restored images (Pokle et al., 2024); in D-Flow, controlled generation is added to an FM or diffusion model, which imposes a prior to the optimization process in an image inverse problem, resulting in better model generalizability (Ben-Hamu et al., 2024); in Flow Priors, the Iterative Corrupted Trajectory Matching (ICTM) algorithm is applied to approximate the maximum-a-posteriori (MAP) estimator in an FM model, which improves the image restoration performance (Zhang et al., 2024); in the latest PnP-Flow, the Plug-and-Play framework is combined with a pre-trained FM, yielding a time-dependent image denoiser with improved computational efficiency and memory consumption (Martin et al., 2025).

In the proposed ECS probability flow learning model, the technical tool for addressing the infinite dimensionality challenge is the moment kernel transform, which is motivated by the method of moments in functional analysis and probability theory (Billingsley, 1995; Yoshida, 1980; Lax, 2002). This method was developed by P. L. Chebyshev in 1887 and formalized by his student A. Markov (Mackey, 1980). It was soon gained considerable attentions and studied under different settings, notably, the Hausdorff, Hamburger, and Stieltjes moment problems (Hausdorff, 1923; Hamburger, 1920; 1921a;b; Stieltjes, 1993). In the moment kernel transform, we adopt the modern mathematical formulation of the moment method, proposed by K. Yosida (Yoshida, 1980).

## 2 REPRESENTATION OF DISTRIBUTION FLOWS USING DYNAMIC ENSEMBLE SYSTEMS

Flow matching (FM) is an emerging machine learning model for targeted transportation of probability distributions in a continuous time. This inherently highlights the close relationship between FM and dynamical systems. In this section, followed by a brief review of FM interpreted through the

lens of dynamical systems, we introduce a new learning model for probability flows, represented by an ensemble of heterogeneous control systems with distinct dynamic characteristics.

## 2.1 FLOW MATCHING FROM A DYNAMICAL SYSTEMS VIEWPOINT

Given a pair of initial and target probability distributions (measures) on a sample space $\mathbb{R}^n$, denoted by $\mu_0$ and $\mu_1$, respectively, the concept of FM involves training a model to generate a predetermined path of probability distributions $\mu_t$, $0 \leq t \leq 1$, which interpolates between $\mu_0$ and $\mu_1$. Specifically, when data sampled from $\mu_0$ is fed into the trained FM model, the output is the temporal evolution of the data, providing a set of samples drawn from $\mu_t$ at each time $t \in [0, 1]$. At the final time $t = 1$, the data distribution reaches the desired distribution $\mu_1$.

The distribution flow learned by FM can also be generated by a dynamical system under some weak regularity conditions. These include the requirement that $\mu_t$ has finite $k^{\text{th}}$ moments for some $k > 1$, i.e., $\int_\Omega |x|^k d\mu_t(x) < \infty$ with $|\cdot|$ being a norm on $\mathbb{R}^d$, for each $t \in [0, 1]$. Additionally, the probability curve $\mu_t$ must be absolutely continuous with an integrable, almost everywhere defined time-derivative on $[0, 1]$. Under these conditions, there exists a vector field $v(t, x) \doteq v_t(x)$ on $\mathbb{R}^n$ such that $\mu_t$ satisfies the continuity equation $\frac{\partial}{\partial t}\mu_t + \nabla \cdot (v_t \mu_t) = 0$ as a weak solution, where $\nabla$ is the gradient operator on $\mathbb{R}^d$ (Ambrosio et al., 2008). This continuity equation is also accompanied by its characteristic ordinary differential equation system on $\mathbb{R}^d$, given by

$$\frac{d}{dt}x(t) = v(t, x(t)). \tag{1}$$

In other words, with data sampled from $\mu_0$ as the initial condition, the time-$t$ solutions of this system form a set of samples drawn from $\mu_t$ (Ambrosio et al., 2008). In this dynamical systems-theoretic context, FM is essentially a model trained to learn the vector field $v : [0, 1] \times \mathbb{R}^d \to \mathbb{R}^n$. Denoting the trainable parameter by $\theta \in \mathbb{R}^r$, which is used to parameterize the learning target, the FM training process can be formulated as

$$\min_{\theta \in \mathbb{R}^r} \ \mathbb{E} \int_0^1 |v(t_i, x(t); \theta) - v(t, x(t))|^2 dt,$$

$$\text{s.t.} \ \frac{d}{dt}x(t) = v(t, x(t); \theta), \ x(0) \sim \mu_0, \tag{2}$$

where $x(0) \sim \mu_0$, i.e., $x(0)$ is a random variable with distribution $\mu_0$, and $\mathbb{E}$ is the expectation with respect to the distribution of the stochastic process $x(t)$.

**Fundamental limits of FM.** The training process shown in (2) falls into the category of an optimal control problem, with the training parameter $\theta$ playing the role of the control input. This control-theoretic interpretation immediately reveals two fundamental limits of FM. Typically, control inputs are time-varying, rather than constant as $\theta$. Restricting feasible control inputs to constant functions results in compromised control performance, which means the learning performance of the FM model is also compromised. On the other hand, the uniqueness of solutions to the characteristic system in (1) adds another layer of limitations on FM. For instance, the transport from the standard normal distribution to the binomial distribution on $\mathbb{R}$ is infeasible, because it requires solution trajectories of the characteristic system starting from different initial conditions to intersect at the point 0 or 1, violating the uniqueness of solutions. This conclusion particularly prevents FM from being used in classification tasks. In general, any pair of distributions with different support cardinalities cannot be transported by FM. To overcome these limitations of FM, we propose a new control system model for learning probability flows.

## 2.2 ENSEMBLE SYSTEMS REPRESENTATION OF PROBABILITY FLOWS

An *ensemble control system (ECS)* is a parameterized family of control systems evolving on a common manifold $M \subseteq \mathbb{R}^n$, described by

$$\frac{d}{dt}x(t, \beta) = v\big(t, \beta, x(t, \beta), u(t)\big), \tag{3}$$

where the system parameter $\beta$ varies over $\Omega \subseteq \mathbb{R}^d$, $x(t, \beta) \in M$ holds for any $t \in [0, 1]$ and $\beta \in \Omega$, $v(t, \beta, \cdot, u(t))$ is a vector field on $M$, and $u(t) \in \mathbb{R}^r$ is the control input. Ensemble

systems are natural models for describing the dynamics of large-scale population systems arising from numerous domains, including robot swarms in robotics, nuclear spin ensembles in quantum control, and spiking neurons in neuroscience (Becker et al., 2013; Zhang & Li, 2015; Li et al., 2013).

A significant difference between the ECS in equation 3 and the characteristic system in (1) in FM is that ECS has heterogeneous dynamics parameterized by $\beta \in \Omega$. To elaborate further on this distinction, starting from a collection of initial conditions $x(0, \beta_i)$ for $i = 1, \ldots, N$, for example, with $\beta_i$ drawn from a probability distribution on $\Omega$, the trajectories $x(\cdot, \beta_i) : [0, 1] \to M$ follow distinct systems for different $\beta_i$. In particular, the non-intersection restriction on different trajectories, as in the case of the characteristic system in FM, is automatically lifted. This, in turn, demonstrates an enhanced capability of the ECS model to represent a greater variety of distribution flows than the FM model.

On the other hand, since the purpose of the control input $u(t)$ is to regulate the system dynamics represented by the vector field $v$, it naturally serves as the training parameter for learning the desired probability flow. In this context, the learning task admits the following optimal tracking control formulation:

$$\min_{u:[0,1]\to\mathbb{R}^r} \quad \mathbb{E} \int_0^1 |v(t, \beta, x(t, \beta), u(t)) - v(t, x(t, \beta))|^2 dt,$$

$$\text{s.t.} \quad \frac{d}{dt} x(t, \beta) = v\big(t, \beta, x(t, \beta), u(t)\big), \ x(0, \cdot) \sim \mu_0, \tag{4}$$

where $\mathbb{E}$ is the expectation with respect to a probability measure $\lambda$ on $\Omega$. Additionally, the space of the training parameters is significantly enlarged from $\mathbb{R}^r$ to $\mathbb{R}^r$-valued functions defined on $[0, 1]$, further enhancing the ECS model capability.

With the introduced heterogeneity parameter $\beta$ varying on $\Omega$, the ensemble state can be viewed as a function $x(t, \cdot) : \Omega \to M$. Equivalently, the ensemble in (3) can be considered as a control system defined on the space of $M$-valued functions over $\Omega$, which is generally infinite-dimensional. Consequently, the training of the ECS model in (4) necessarily operates on an infinite-dimensional function space, presenting significant challenges from both theoretical and computational perspectives. To address this issue, we develop a new learning representation for probability flows by mapping the ECS model to a reproducing kernel Hilbert space (RKHS), which is the main focus of the next section.

## 3 MOMENT KERNELIZATION FOR DISTRIBUTION FLOW MATCHING

In this section, we develop a moment kernel representation of the ECS model for learning probability flows, addressing the challenge arising from the infinite dimensionality of the model. This approach is enabled by the moment kernel transform, which maps an ensemble system to an associated moment system defined on an RKHS. This moment system gives rise to a reduced kernel representation of the probability flow generated by the ensemble system, thereby enabling efficient probability flow learning within the moment domain.

### 3.1 MOMENT KERNEL TRANSFORM

The development of the moment kernel transform is inspired by the method of moments in functional analysis (Yoshida, 1980), which involves representing Borel measures using sequences. In the context of this work, we focus on Radon probability measures on $\Omega$ and denote the space of such measures by $\mathcal{P}(\Omega)$. It follows from the renowned Riesz–Markov–Kakutani representation theorem that every positive linear functional on $C_c(\Omega)$, the space of compactly supported continuous real-valued functions defined on $\Omega$, is uniquely representable as a measure on $\mathcal{P}(\Omega)$ (Folland, 1999). This allows us to introduce the notion of *moments* for $\mu_t \in \mathcal{P}(\Omega)$ by using the dual-primal pairing $\langle \cdot, \cdot \rangle : C_c(\Omega) \times \mathcal{P}(\Omega) \to \mathbb{R}$ as

$$m_k(t) = \langle \varphi_k, \mu_t \rangle = \int_\Omega \varphi_k d\mu_t, \quad k \in \mathbb{N}. \tag{5}$$

Here, $\{\varphi_k\}_{k\in\mathbb{N}}$ is chosen to be a basis for $C_c(\Omega)$, which guarantees a one-to-one correspondence between the measure $\mu_t$ and its *moment sequence* $m(t) = \big(m_k(t)\big)_{k\in\mathbb{N}}$ (Yoshida, 1980).

**The RKHS structure on the moment space.** To incorporate the moment kernel transform into the ECS framework, we focus on Borel measures on $\Omega$ that are absolutely continuous with respect to a probability measure $\lambda \in \mathcal{P}(\Omega)$. The Radon-Nikodym derivative of these measures is given by the ensemble state $x_t$, i.e., $d\mu_t = x_t d\lambda$. In this case, the dual-primal pairing can be identified with the $L^2$-inner product of the functions $\varphi_k$ and $x_t$ with respect to the measure $\lambda$ as $m_k(t) = \langle \varphi_k, \mu_t \rangle = \int_\Omega \varphi_k x_t d\lambda = \langle \varphi_k, x_t \rangle_{L^2}$, provided $\|x_t\|^2 = \int_\Omega |x_t|^2 d\lambda < \infty$, i.e., $x_t \in L^2(\Omega, \lambda)$. This identification further allows the choice of $\{\varphi_k\}_{k \in \mathbb{N}}$ as an orthonormal basis of $L^2(\Omega, \lambda)$, in which case the moment sequence $m(t)$ satisfies $\|m(t)\|_{\ell^2} = \sum_{k \in \mathbb{N}} |m_k(t)|^2 = \|x_t\|^2_{L^2} < \infty$, indicating that $m(t)$ is an element in $\ell^2$, the space of square-summable sequences (Folland, 1999). It has been shown that the $\ell^2$-space is an RKHS with the reproducing kernel given by the inner product $\langle m(t_1), m(t_2) \rangle_{\ell^2} = \sum_{k \in \mathbb{N}} m_k(t_1) m_k(t_2)$ (Paulsen & Raghupathi, 2016). This demonstrates that $m(t)$ is essentially a kernel representation of $\mu_t$, which will be leveraged to efficiently facilitate ensemble control-based probability flow learning.

### 3.2 MOMENT KERNEL REPRESENTATION OF ENSEMBLE FLOW MATCHING

The crucial step in kernelizing the probability flow learning problem in (4) is to derive the moment kernel representation of the ensemble system. To this end, we differentiate $m_k(t)$, defined in (5), with respect to $t$, yielding

$$\frac{d}{dt} m_k(t) = \frac{d}{dt} \langle \varphi_k, \mu_t \rangle = \frac{d}{dt} \int_\Omega \varphi_k x_t d\lambda = \int_\Omega \varphi_k \frac{d}{dt} x_t d\lambda = \int_\Omega \varphi_k v(t, \cdot, x_t, u(t)) d\lambda,$$

where the interchange of integration and differentiation follows from the dominated convergence theorem (Folland, 1999). Notice that the last term above is essentially the $k^{\text{th}}$ moment of the $x_t$-dependent measure whose Radon-Nikodym derivative with respect to $\lambda$ is $v(t, \beta, x_t(\beta), u(t))$ as a function of $\beta$. Representing the moment sequence of $v$ as $\bar{v} = (\bar{v}_k)_{k \in \mathbb{N}}$, the moment kernel representation of the ensemble system follows $\frac{d}{dt} m(t) = \bar{v}(t, m(t), u(t))$. On the other hand, the training loss in (4) satisfies $\mathbb{E} \int_0^1 |v(t, \beta, x(t, \beta), u(t)) - v(t, x(t, \beta))|^2 dt = \int_\Omega \int_0^1 |v(t, \beta, x(t, \beta), u(t)) - v(t, x(t, \beta))|^2 dt d\lambda(\beta) = \int_0^1 \int_\Omega |v(t, \beta, x(t, \beta), u(t)) - v(t, x(t, \beta))|^2 d\lambda(\beta) dt = \int_0^1 \|v(t, \cdot, x_t, u(t)) - v(t, x_t)\|^2_{L^2} dt = \int_0^1 \|\bar{v}(t, m(t), u(t)) - \bar{v}(t, m(t))\|^2_{\ell^2} dt$, where the change in the order of integration follows from Tonelli's theorem (Folland, 1999). We have now reached the moment kernel representation of the ensemble flow matching in (4), given by

$$\min_{u:[0,1] \to \mathbb{R}^r} \int_0^1 \|\bar{v}(t, m(t), u(t)) - \bar{v}(t, m(t))\|^2 dt,$$

$$\text{s.t.} \quad \frac{d}{dt} m(t) = \bar{v}(t, m(t), u(t)), \; m(0) = \Big( \langle \varphi_k, \mu_0 \rangle \Big)_{k \in \mathbb{N}}. \tag{6}$$

**Moment kernelized ECS model training.** The sequential nature of the moment kernel representation suggests an effective training approach to the model in (4) by using a finite truncation of the moment sequence $m(t)$. Let $\hat{m}_N = (m_0, m_1, \ldots, m_N)$ and $\hat{v}_N = (v_0, v_1, \ldots, v_N)$ denote the respective order-$N$ truncated moment sequences for $m$ and $v$. We can then construct a finite-dimensional approximation of the moment kernelized ensemble FM model in (6) as

$$\min_{u:[0,1] \to \mathbb{R}^r} \int_0^1 \|\hat{v}_N(t, \hat{m}_N(t), u(t)) - \hat{v}_N(t, \hat{m}_N(t))\|^2 dt,$$

$$\text{s.t.} \quad \frac{d}{dt} \hat{m}_N(t) = \hat{v}_N(t, \hat{m}_N(t), u(t)), \; \hat{m}_N(0) = \Big( \langle \varphi_k, \mu_0 \rangle \Big)_{k=0}^N. \tag{7}$$

**Theorem 1** *Let $m^*(t)$ and $\hat{m}_N^*(t)$ be the moment kernelized probability flows learned by training the infinite-dimensional and truncated models in (6) and (7), respectively. Then, $\hat{m}_N^*(t) \to m^*(t)$ uniformly in $t \in [0, 1]$ as the truncation order $N \to \infty$.*

*Proof.* See Appendix A. □

On the other hand, the values of the moments can be revealed from the data with little efforts. Leveraging the fact that $\mu_t$ is a probability measure on $\Omega$, the $k^{\text{th}}$ moment defined in (5) is essentially

the expectation of the random variable $\varphi_k$ with respect to $\mu_k$, i.e., $m_k(t) = \mathbb{E}_{\mu_t}(\varphi_k)$. Therefore, $m_k(t)$ can be approximated using the sample moment of the data sampled from $\mu_t$, by the law of large numbers. Moreover, because the components of $\bar{v}$ are also the moments of $v$, they also admit the sample moment approximation.

# 4 DYNAMIC IMAGE PROCESSING THROUGH CONTROLLING ENSEMBLE DISTRIBUTION FLOWS

One of the primary application domains of FM is image processing. In this section, we will apply the developed ensemble system model to address various image processing tasks, including denoising, undistortion, and undegradation, using benchmark datasets. In addition, we will conduct a detailed comparison with existing FM models to assess performance differences.

## 4.1 EXPERIMENTAL EVALUATION

**Datasets.** We validate the proposed ECS model using three benchmark datasets, including CelebA-HQ and AFHQ-Cat, and LSUN-bedroom, which contain RGB images with resolutions of $128 \times 128$, $256 \times 256$, and $128 \times 128$, respectively (Yang et al., 2015; Choi et al., 2020; Yu et al., 2015). For each image in CelebA-HQ, we reshape the data into three $128^2$-dimensional vectors, each corresponding to one color channel. Each vector is then normalized to a (uniformly discretized) probability density function over the interval $[-1, 1]$ for the ESC model to process. Images in AFHQ-Cat are processed in the same manner.

**ECS model setup.** It is widely known that images are typically processed using convolution operations, which are linear transforms (Wiatowski & Bölcskei, 2018; Gonzalez & Woods, 2006). This inspires the use of a linear ensemble system to process the "image-distributions." Specifically, we build the ensemble system model with linear variation in its natural dynamics, controlled by multiple parameter-dependent inputs, in the form

$$\frac{d}{dt}x(t, \beta) = \beta x(t, \beta) + \sum_{i=0}^{q} \varphi_i(\beta) u_i(t),$$

where $\varphi_i$ is the order-$i$ Legendre polynomial. Furthermore, we utilize the set of Legendre polynomials, which is an orthonormal basis for $L^2([-1, 1])$ with respect to the Lebesgue measure, to define the moments $m_k(t) = \langle \varphi_k, \mu_t \rangle = \int_{-1}^{1} \varphi_k(\beta) d\mu_t(\beta) = \int_{-1}^{1} \varphi_k(\beta) x_t(\beta) d\beta$ for all $k \in \mathbb{N}$. The initial and target distributions, $\mu_0$ and $\mu_1$, correspond to the degraded and ground truth images, respectively. The probability path to match is chosen to be the linear interpolation $\mu_t = (1-t)\mu_0 + t\mu_t$, $t \in [0, 1]$ between them (see Appendix B for details).

**ECS model Hyperparameter Selection.** This linear ECS model has two hyperparameters: the number $q$ of control inputs and the order $p$ of the truncation for the moment kernelization. To ensure that the ECS model has the ability to track the desired probability flow, We set $p = q$ (see Appendix B.2 for control analysis of the linear ECS model capability). Specifically, $p$ and $q$ are chosen to be 150 for images in the CelebA-HQ dataset and 250 for images in the AFHQ-Cat dataset.

**Baseline models.** We compare our model with four state-of-the-art FM-based image processing models, including OT-ODE (Pokle et al., 2024), D-Flow (Ben-Hamu et al., 2024), Flow-Priors (Zhang et al., 2024), and PnP-Flow (Martin et al., 2025). The performance measures used for comparison are the peak signal-to-noise ratio (PSNR) and the structural similarity index (SSIM), with higher values indicating better performance.

## 4.2 SIMULATION RESULTS

To showcase the performance of the proposed ECS model, we consider four different image restoration tasks: denoising, deblurring, super-resolution, and inpainting. For each task, we randomly select 100 images from the CelebA-HD database. (1) Denoising: The selected images are perturbed by Gaussian noise with a standard deviation of 0.2; (2) Deblurring: two different blurring methods are

applied to the images: (i) the Gaussian blur with a $61 \times 61$ Gaussian kernel with a standard deviation of 20 and (ii) the motion blur with a length of 45 (pixels) and an angle of $45°$; (3) Super-resolution: The images are downsampled by a factor of 4; and (4) Inpainting: Two inpainting methods are considered: (i) single-box inpainting using a mask of size $50 \times 50$ pixels at the center and (ii) multi-box impainting using 100 randomly placed $5 \times 5$-pixel masks. In addition to the aforementioned perturbations, we also add extra Gaussian noise with a standard deviation of 0.01 to all the perturbed images as measurement noise.

**ECS model evaluation.** Figure 1 illustrates the restoration performance of the ECS model, where we select one image to showcase the performance for each task. Specifically, the first and second columns represent the ground truth and degraded images, respectively. The third to sixth columns display the outputs of the ESC model at time points 0.25, 0.5, 0.75, and 1, respectively. We observe that the restored images are almost identical to the original ones, supported by the high PSNR values. Additional experimental results are provided in Appendix C, where we also use various experiments to show that the ECS model has excellent noise robustness and an outstanding ability to tackle high-resolution images.

**Comparison with baseline models.** To conduct a fair comparison with the baseline models, we slightly modify the multi-box inpainting and downsampling procedures as follows: the 100 $5 \times 5$-pixel randomly placed inpainting boxes are replaced by 70% randomly selected inpainting pixels, and the downsampling factor is decreased from 4 to 2. For each image restoration task, the model performance is evaluated using the average PSNR and SSIM over the 100 selected images. The comparison results for the CelebA-HQ and AFHQ-Cat databases are shown in Tables 1 and 2, respectively. Moreover, Figure 2 displays one image restored by each model for each task. We observe that the proposed ECS model either outperforms or performs as well as the baseline models, as indicated by the PSNR values.

Table 1: Comparison between the proposed ECS model and baseline models using the CelebA-HD database for image denoising, deblurring, inpainting of single and multiple masks, and super-resolution.

| Method | Denoising ($\sigma = 0.2$) | | Deblurring ($61 \times 61$, $\sigma_b = 1$) | | Box inpaint. ($40 \times 40$) | | Rand. inpaint. (70%) | | Super-res. ($2\times$) | |
|---|---|---|---|---|---|---|---|---|---|---|
| | PSNR | SSIM | PSNR | SSIM | PSNR | SSIM | PSNR | SSIM | PSNR | SSIM |
| Degraded | 20.00 | 0.348 | 27.67 | 0.740 | 22.12 | 0.742 | 11.82 | 0.197 | 10.17 | 0.182 |
| OT-ODE | 30.50 | 0.867 | 32.63 | 0.915 | 28.84 | 0.914 | 28.36 | 0.865 | 31.05 | 0.902 |
| D-Flow | 26.42 | 0.651 | 31.07 | 0.877 | 29.70 | 0.893 | 33.07 | 0.938 | 30.75 | 0.866 |
| Flow-Priors | 29.26 | 0.766 | 31.40 | 0.856 | 29.40 | 0.858 | 32.33 | 0.945 | 28.35 | 0.717 |
| PnP-Flow | 32.45 | 0.911 | 34.51 | 0.940 | 30.59 | 0.943 | 33.54 | 0.953 | 31.49 | 0.907 |
| Ensemble Control (ours) | 33.47 | 0.964 | 34.48 | 0.964 | 31.27 | 0.958 | 33.25 | 0.957 | 30.94 | 0.900 |

Table 2: Comparison between the proposed ECS model and baseline models using the AFHQ-Cat database for image denoising, deblurring, inpainting of a single mask and multiple masks, and super-resolution.

| Method | Denoising ($\sigma = 0.2$) | | Deblurring ($\sigma = 0.05$, $\sigma_b = 3.0$) | | Box inpaint. ($80 \times 80$) | | Rand. inpaint. (70%) | | Super-res. ($4\times$) | |
|---|---|---|---|---|---|---|---|---|---|---|
| | PSNR | SSIM | PSNR | SSIM | PSNR | SSIM | PSNR | SSIM | PSNR | SSIM |
| Degraded | 20.00 | 0.319 | 23.77 | 0.514 | 21.50 | 0.744 | 13.35 | 0.234 | 11.59 | 0.216 |
| OT-ODE | 29.90 | 0.831 | 26.43 | 0.709 | 23.88 | 0.874 | 28.84 | 0.838 | 25.17 | 0.711 |
| D-Flow | 26.22 | 0.620 | 27.49 | 0.740 | 26.69 | 0.833 | 31.37 | 0.888 | 24.10 | 0.595 |
| Flow-Priors | 29.32 | 0.768 | 25.78 | 0.692 | 25.85 | 0.822 | 31.76 | 0.909 | 23.34 | 0.573 |
| PnP-Flow | 31.65 | 0.876 | 27.62 | 0.763 | 26.87 | 0.904 | 32.98 | 0.930 | 26.75 | 0.774 |
| Ensemble Control (ours) | 32.95 | 0.917 | 29.91 | 0.867 | 30.09 | 0.927 | 32.03 | 0.943 | 27.33 | 0.785 |

## 4.3 DISCUSSION ON THE ECS MODEL

In addition to the model capability discussed in Section 2.2, the interpretability and high training efficiency are other major advantages of the ECS model. As a control system representation learning model, its trainable parameters are the control inputs of the corresponding ensemble control system, which guide the system dynamics as desired. On the other hand, training the ECS model is as straightforward as solving a time-varying least-squares problem, as demonstrated in Appendix

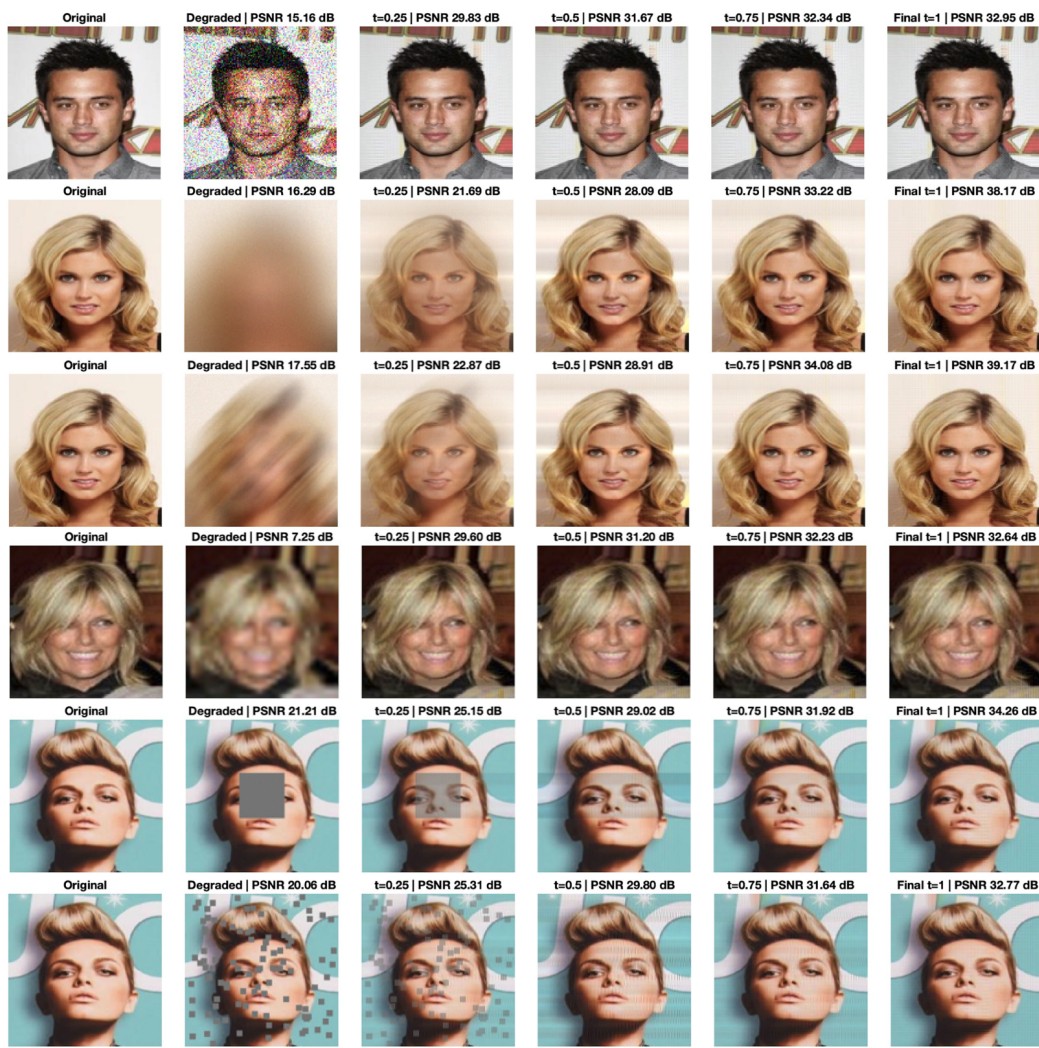

Figure 1: Illustration of image restoration processes generated by the ECS model, including denoising (row 1), deblurring of Gaussian blur (row 2) and motion blur (row 3), super-resolution (row 4), and inpainting of a single mask (row 5) and multiple masks (row 6). Columns 1 and 2 are the ground truth and degraded images; columns 3 to 6 display the output images, with their PSNR values, of the ECS model at time points 0.25, 0.5, 0.75, and 1.

B.2. This approach does not require any time-consuming pre-training of deep neural networks. For example, the total time required for training and processing the CelabA image deblurring tasks is 16.07 seconds (average per image) on an Apple M4 Max chip with 128GB RAM. This is comparable to the processing times solely of baseline models running on a more advanced NVIDIA RTX 6000 Ada Generation with 48GB RAM: 1.50 seconds for OT-ODE, 16.01 seconds for Flow-Priors, 32.19 seconds for D-Flow, and 3.40 seconds for PnP-Flow (Martin et al., 2025).

It is worth pointing out that the distribution flows generated by the ECS model are not necessarily solutions of continuity equations. This indicates that ECS has the ability to learn and generate non-probability distribution flows. Indeed, when the ensemble state $x_t$ is a general $L^2$-function, rather than a probability density, the measure $\mu_t$ becomes a signed measure, which can take negative values on some subsets of $\Omega$. This characteristic allows ECS to handle a broader range of applications beyond traditional probability distribution flows.

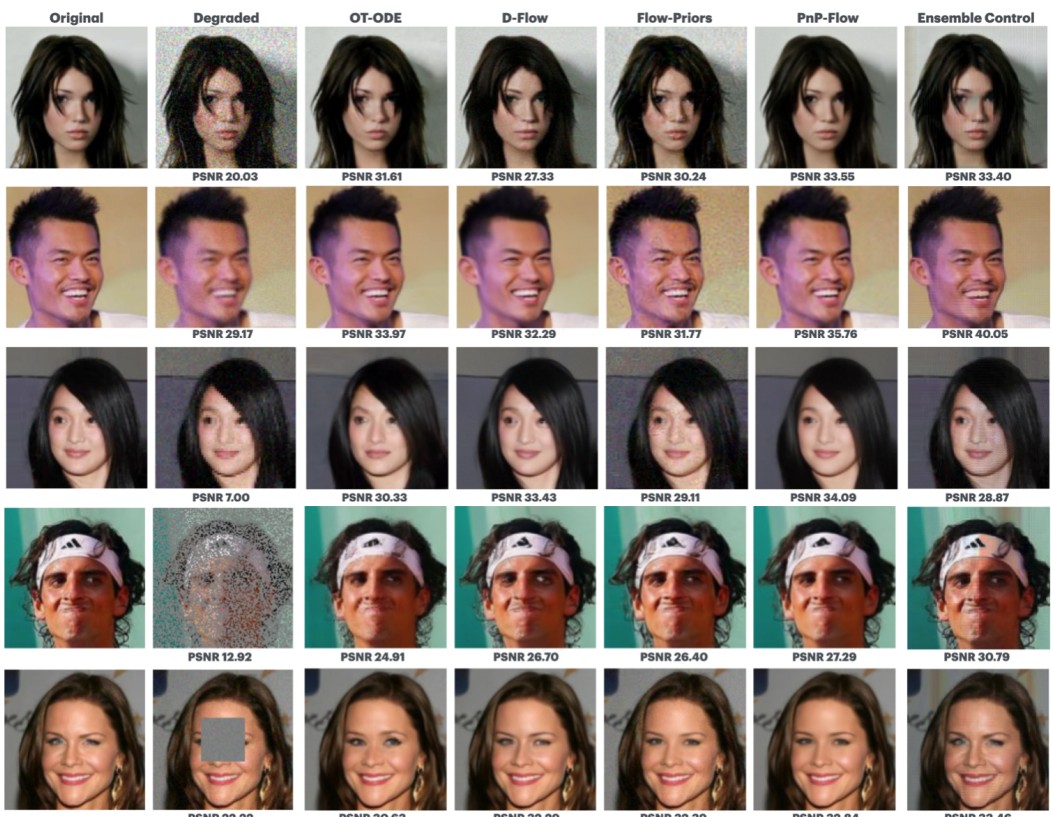

Figure 2: Illustration of the comparison results between the proposed ECS model and baseline models for imaging denoising (row 1), deblurring (row 2), super-resolution (row 3), inpainting of multiple masks (row 4) and a single mask (row 5). Columns 1 and 2 are the ground truth and degraded images; columns 3 to 6 are the restored images output from the baseline models, including OT-ODE, D-Flow, Flow-Priors, and PnP-Flow; column 7 display the images restored by the ECS model.

## 5 CONCLUSION

In this paper, inspired by an in-depth exploration of the dynamical systems aspect of flow matching (FM), we propose an ensemble control system (ECS) representation model for learning probability flows. In the ECS model, probability flows are generated by an ensemble of heterogeneous control systems, rather than a single unforced dynamical system, with the control inputs serving as time-varying trainable parameters. The heterogeneous dynamics and time-dependent trainable parameters greatly enhance the model capability of ECS, making it exceptionally powerful. To further capitalize on these strengths, we develop a moment kernel transform, which generates a reduced kernel representation of ECS over an RKHS providing efficient training. We demonstrated these significant advantages of the ECS model through various image restoration tasks and provided a detailed comparison with baseline FM-based image processing models.

**Limitations.** As mentioned at the end of Section 3.2, the moment kernelized ECS model is approximated in terms of sample moments. While the law of large numbers provides a theoretical guarantee that the sample moments converge to the true moments, in practice, this requires a sufficient amount of data to achieve an accurate approximation. Without adequate data, the learning performance of the ECS model may be compromised, which is a fundamental limitation for general learning algorithms.

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

## A    PROOF OF THEOREM 1

*Assumption.* The vector field $v(t, \beta, x_t, u(t))$, as a function in $\beta$, is an $L^2$-function on $\Omega$, which is Lipchitz continuous in $(t, x_t)$ uniformly for $u(t)$. Meaning, there exists $L < \infty$, independent of $u$, such that $\|v(t_1, \cdot, x_{t_1}, u(t_1)) - v(t_1, \cdot, x_{t_1}, u(t_1))\|_{L^2} \leq L\big(|t_1 - t_2| + \|x_{t_1} - x_{t_2}\|_{L^2}\big)$.

This is the basic condition in the theory of ordinary differential equations to guarantee that, driven by any control input $u(t)$, the ECS model has a unique solution $x_t$ (Bardi & Capuzzo-Dolcetta, 2009). Because the moment kernel transform $L^2(\Omega) \to \ell^2$ given by $x_t \mapsto m(t)$, defined in Section 3.1, is an isometry, the vector field $\bar{v}$ in the moment kernelized ECS model also satisfies the Lipschitz continuity condition with the same Lipschitz constant $L$. Moreover, as a projection on to the first $N$ components of $\bar{v}$, this assumption holds for the vector field $\hat{v}_N$ in the truncated moment kernelized ECS model as well for all $N$.

*Proof of Theorem 1.* It suffices to prove that, with a fixed trainable control function $u(t)$, the output $\hat{m}_N(t)$ of the truncated moment kernelized ESC model in (7) converges to that $m(t)$ of the entire moment kernelized ESC model in (6).

We first show $\hat{v}_N \to \bar{v}$ as $N \to \infty$. As pointed out in Section 3.2, $\bar{v}(t, m(t), u(t)) \in \ell^2$ is the moment sequence of $v(t, \cdot, x_t, u(t)) \in L^2(\Omega)$ and $\hat{v}_N(t, m(t), u(t))$ is the order-$N$ truncation of $v(t, m(t), u(t))$, which yields $\|\bar{v}(t, m(t), u(t)) - \hat{v}_N(t, m(t), u(t))\|_{\ell^2} \to 0$ as $N \to \infty$. The Lipschitz continuity of $\bar{v}$ and $\hat{v}_N$ particularly implies their continuity, and hence we also have $\|\bar{v}(t, m(t), u(t)) - \bar{v}(t, \hat{m}_N(t), u(t))\| \to 0$ and $\|\hat{f}_N(t, m(t), u(t)) - \hat{f}_N(t, \hat{m}_N(t), u(t))\|_{\ell^2} \to 0$ as a result of $\|m(t) - \hat{m}_N(t)\|_{\ell^2} \to 0$ as $N \to \infty$ (Apostol, 1974). This vector field convergence then follows from the triangle inequality as

$$\|\bar{v}(t, m(t), u(t)) - \hat{v}_N(t, \hat{m}_N(t), u(t))\|_{\ell^2} \leq \|\bar{v}(t, m(t), u(t)) - \hat{v}_N(t, m(t), u(t))\|_{\ell^2}$$
$$+ \|\hat{v}_N(t, m(t), u(t)) - \hat{v}_N(t, \hat{m}_N(t), u(t))\|_{\ell^2} \to 0$$

as $N \to \infty$.

The convergence of the vector fields will be now adopted to prove the desired convergence for the trajectories of the moment and truncated moment kernelized ECS models. To avoid possible confusion about the moment order and truncation order, in the following, we use superscript to denote the moment order, e.g., $m(t) = \left(m^k(t)\right)_{k \in \mathbb{N}}$. Then, we have both of

$$\frac{d}{dt}\left(m^k(t) - \hat{m}_N^k(t)\right) = \bar{v}^k(t, m(t), u(t)) - \hat{v}_N^k(t, \hat{m}_N(t), u(t))$$
$$\leq \left|\bar{v}^k(t, m(t), u(t)) - \hat{v}_N^k(t, \hat{m}_N(t), u(t))\right|$$

and

$$\frac{d}{dt}\left(m_N^k(t) - \hat{m}^k(t)\right) = \bar{v}_N^k(t, \hat{m}_N(t), u(t)) - \hat{v}^k(t, m(t), u(t))$$
$$\leq \left|\bar{v}^k(t, m(t), u(t)) - \hat{v}_N^k(t, \hat{m}_N(t), u(t))\right|,$$

which yield

$$\frac{d}{dt}\left|m^k(t) - \hat{m}_N^k(t)\right| \leq \left|\bar{v}^k(t, m(t), u(t)) - \hat{v}_N^k(t, \hat{m}_N(t), u(t))\right|.$$

Then, the monotone convergence theorem implies

$$\frac{d}{dt}\|m(t) - \hat{m}_N(t)\|_{\ell^2}^2 = \frac{d}{dt}\sum_{k=0}^{\infty}\left|m^k(t) - \hat{m}_N^k(t)\right|^2 = \sum_{k=0}^{\infty}\left|m^k(t) - \hat{m}_N^k(t)\right|\frac{d}{dt}\left|m^k(t) - \hat{m}_N^k(t)\right|$$

$$\leq \sum_{k=0}^{\infty}\left|m^k(t) - \hat{m}_N^k(t)\right|\left|\bar{v}^k(t, m(t), u(t)) - \hat{v}_N^k(t, \hat{m}_N(t), u(t))\right|$$

$$\leq \|m(t) - \hat{m}_N(t)\|_{\ell^2}\|\bar{v}(t, m(t), u(t)) - \hat{v}_N(t, \hat{m}_N(t), u(t))\|_{\ell^2}$$

$$\leq \|m(t) - \hat{m}_N(t)\|_{\ell^2}\Big(\|\bar{v}(t, m(t), u(t)) - \hat{v}_N(t, m(t), u(t))\|_{\ell^2}$$

$$+ \|\hat{v}_N(t, m(t), u(t)) - \hat{v}_N(t, \hat{m}_N(t), u(t))\|_{\ell^2}\Big).$$

By using $\frac{d}{dt}\|m(t) - \hat{m}_N(t)\|_{\ell^2}^2 = \|m(t) - \hat{m}_N(t)\|_{\ell^2}\frac{d}{dt}\|m(t) - \hat{m}_N(t)\|_{\ell^2}$, we obtain

$$\frac{d}{dt}\|m(t) - \hat{m}_N(t)\| \leq \|\bar{v}(t, m(t), u(t)) - \hat{v}_N(t, m(t), u(t))\|_{\ell^2}$$

$$+ \|\hat{v}_N(t, m(t), u(t)) - \hat{v}_N(t, \hat{m}_N(t), u(t))\|_{\ell^2}$$

$$\leq \|\bar{v}(t, m(t), u(t)) - \hat{v}_N(t, m(t), u(t))\|_{\ell^2} + L\|m(t) - \hat{m}_N(t)\|_{\ell^2},$$

where we use the Lipschitz continuity of $\hat{v}_N$. Because of the convergence of $\hat{v}_N$ to $\bar{v}$, for any $\varepsilon > 0$, we can choose a large enough $N$ such that $\|\bar{v}(t, m(t), u(t)) - \hat{v}_N(t, m(t), u(t))\|_{\ell^2} < \varepsilon$, which gives

$$\frac{d}{dt}\|m(t) - \hat{m}_N(t)\| \leq L\|m(t) - \hat{m}_N(t)\|_{\ell^2} + \varepsilon.$$

By Grönwall's inequality (Bardi & Capuzzo-Dolcetta, 2009), the above estimate yields an upper bound for the cumulative truncation error, given by,

$$\|m(t) - \hat{m}_N(t)\|_{\ell^2} \leq e^{Lt}\Big(\|m(0) - \widehat{m}_N(0)\|_{\ell^2} + \varepsilon t\Big) \leq e^L\Big(\|m(0) - m_N(0)\|_{\ell^2} + \varepsilon\Big)$$

for $t \in [0, 1]$. Because $\hat{m}_N(0)$ is chosen to be the order-$N$ truncation of $m(t)$, we have $\|m(0) - m_N(0)\|_{\ell^2}$. Then, letting $\varepsilon \to 0$, we obtain $\sup_{t \in [0,1]}\|m(t) - \hat{m}_N(t)\|_{\ell^2} \to 0$ as $N \to \infty$, concluding the proof. $\qquad\square$

## B  ANALYSIS OF THE LINEAR ECS MODEL

This section is devoted to a detailed analysis of the linear ECS model for image restoration tasks, that is,

$$\frac{d}{dt}x(t, \beta) = \beta x(t, \beta) + \sum_{i=0}^{q}\varphi_i(\beta)u_i(t), \tag{8}$$

where $\beta \in [-1, 1]$ and $\varphi_i$ is the order-$i$ Legendre polynomial. Specifically, we will explicitly derive the moment kernel representation of this ECS model and then analyze its model capability through controllability of the moment kernelized system.

## B.1 MOMENT KERNEL REPRESENTATION OF THE LINEAR ECS

In addition to the orthonormality, the set of Legendre polynomials $\{\varphi_k\}_{k\in\mathbb{N}}$ satisfies the Bonnet's recursion formula, given by,

$$\frac{2k+1}{\sqrt{2k+1}}\beta\varphi_k(\beta) = \frac{k}{\sqrt{2k-1}}\varphi_{k-1}(\beta) + \frac{k+1}{\sqrt{2k+3}}\varphi_{k+1}(\beta)$$

for all $k \in \mathbb{N}$ with $\varphi_{-1}$ defined to be the 0 function (Deift, 2000). Leveraging this relation, we derive the moment kernelized ECS model as

$$\frac{d}{dt}m_k(t) = \frac{d}{dt}\langle\varphi_k, \mu_t\rangle = \frac{d}{dt}\int_{-1}^{1}\varphi_k(\beta)x_t(\beta)d\beta = \int_{-1}^{1}\varphi_k(\beta)\frac{d}{dt}x_t(\beta)d\beta$$

$$= \int_{-1}^{1}\varphi_k(\beta)\Big(\beta x_t(\beta) + \sum_{i=0}^{q}\varphi_i(\beta)u_i(t)\Big)d\beta = \int_{-1}^{1}\beta\varphi_k(\beta)x_t(\beta)d\beta + \sum_{i=0}^{q}\int_{-1}^{1}\varphi_k(\beta)\varphi_i(\beta)u_i(t)d\beta$$

$$= \int_{-1}^{1}\left(\frac{k\varphi_{k-1}(\beta)x_t(\beta)}{\sqrt{(2k-1)(2k+1)}} + \frac{(k+1)\varphi_{2k+1}(\beta)x_t(\beta)}{\sqrt{(2k+1)(2k+3)}}\right)d\beta + \sum_{i=0}^{q}u_i(t)\int_{-1}^{1}\varphi_k(\beta)\varphi_i(\beta)d\beta$$

$$= \frac{k}{\sqrt{(2k-1)(2k+1)}}m_{k-1}(t) + \frac{k+1}{\sqrt{(2k+1)(2k+3)}}m_{k+1}(t) + \sum_{i=0}^{q}\delta_{ki}u_i(t),$$

where $\delta_{ki}$ is the Dirac delta function, that is, $\delta_{ki} = 1$ if $k = i$ and $\delta_{ki} = 0$ otherwise. Represented in the "infinite-dimensional matrix" form, the moment kernelized linear ECS model is given by

$$\frac{d}{dt}\begin{bmatrix} m_0(t) \\ m_1(t) \\ m_2(t) \\ \vdots \\ m_q(t) \\ m_{q+1}(t) \\ \vdots \end{bmatrix} = \begin{bmatrix} 0 & c_1 & 0 & & & & \\ c_0 & 0 & c_2 & & & & \\ 0 & c_1 & 0 & \ddots & & & \\ & \ddots & \ddots & \ddots & & & \\ & & & c_{q-1} & 0 & c_{q+1} & \\ & & & & c_q & 0 & c_{q+2} \\ & & & & & \ddots & \ddots & \ddots \end{bmatrix}\begin{bmatrix} m_0(t) \\ m_1(t) \\ m_2(t) \\ \vdots \\ m_q(t) \\ m_{q+1}(t) \\ \vdots \end{bmatrix}$$

$$+ \begin{bmatrix} 1 & & \\ & \ddots & \\ & & 1 \\ 0 & \cdots & 0 \\ \vdots & & \vdots \end{bmatrix}\begin{bmatrix} u_0(t) \\ \vdots \\ u_q(t) \end{bmatrix}$$

$$= Am(t) + Bu(t),$$

where $c_k = (k+1)/\sqrt{(2k+1)(2k+3)}$. In this moment kernelizd linear ECS model, the banded structure of the system matrix $A$ and the "diagonal" structure of the control matrix $B$ are exactly the manifestations of the Bonnet's recursion formula and the orthonormal property for the Legendre polynomial basis, respectively. It is also worth noting that the transformation from the the linear ECS model to this moment kernelized model leaves the trainable parameters (control inputs) untouched, which gives another verification of the equivalence between training the ECS and moment models.

Of course, the training process can only operate on the truncated model, which we denoted as

$$\frac{d}{dt}\hat{m}_N(t) = \hat{A}_N\hat{m}_N(t) + \hat{B}_Nu(t). \tag{9}$$

Here,

$$\hat{A}_N = \begin{bmatrix} 0 & c_1 & 0 & & \\ c_0 & 0 & c_2 & & \\ & \ddots & \ddots & \ddots & \\ & & c_{q-2} & 0 & c_q \\ & & & c_{q-1} & 0 \end{bmatrix}$$

is the $q \times q$ submatrix of $A$ consisting of the first $N$ rows and columns, while the form of $\hat{B}$ is dependent on the relation between the number of trainable parameters $q$ and the truncation order $N$. In particular, we have

$$
\hat{B}_N = \begin{bmatrix} 1 & & & 0 & \cdots & 0 \\ & \ddots & & \vdots & & \vdots \\ & & 1 & 0 & \cdots & 0 \end{bmatrix} \quad \text{if} \quad N \leq q \quad \text{and} \quad \hat{B}_N = \begin{bmatrix} 1 & & \\ & \ddots & \\ & & 1 \\ \hline 0 & \cdots & 0 \\ \vdots & & \vdots \\ 0 & \cdots & 0 \end{bmatrix} \quad \text{if} \quad N > q.
$$

Intuitively, following from the convergence of $\hat{m}_N(t)$ to $m(t)$, higher-order truncated models should achieve better learning performance. However, as we will discuss in the sequel, higher-order truncated models require more trainable parameters to reach their model capabilities. In another word, the limited number $p$ of control inputs may make higher-order models perform worse then lower-order models. Indeed, we will use control-theoretic techniques to show that $N = p$ is the most effective choice of the truncation order.

## B.2 MODEL CAPABILITY OF THE LINEAR ESC

Conceptually, the model capability characterizes how well a trained machine learning model performs the assigned task. In the context of the truncated moment kernelized ECS model in (9), we are concerned with whether the control parameters $u(t)$ can be tuned so that the model output $\hat{m}_N(t)$ satisfies $\hat{m}_N(t) = \hat{m}_N^*(t)$, where $\hat{m}_N^*(t)$ is the order-$N$ truncated moment kernel representation of the desired probability flow $\mu_t^*$. In the terminology of control theory, this notion of the model capability coincides with path controllability of this order-$N$ kernelized system in (9). Fortunately, the necessary and sufficient path controllability condition for a time-invariant linear control system as this truncated moment has been established in 1965, that is, the $N^2 \times (2qN - q)$ matrix

$$
\begin{bmatrix} \hat{B}_N & \hat{A}_N \hat{B}_N & \hat{A}_N^2 \hat{B}_N & \cdots & \hat{A}_N^{N-1} \hat{B}_N & \hat{A}_N^N \hat{B}_N & \cdots & \hat{A}_N^{2N-1} \hat{B}_N \\ & \hat{B}_N & \hat{A}_N \hat{B}_N & \cdots & \hat{A}_N^{N-2} \hat{B}_N & \hat{A}_N^{N-1} \hat{B}_N & \cdots & \hat{A}_N^{2N-2} \hat{B}_N \\ & & \hat{B}_N & \cdots & \hat{A}_N^{N-3} \hat{B}_N & \hat{A}_N^{N-2} \hat{B}_N & \cdots & \hat{A}_N^{2N-3} \hat{B}_N \\ & & & \ddots & \vdots & \vdots & & \vdots \\ & & & & \hat{B}_N & \hat{A}_N \hat{B}_N & \cdots & \hat{A}_N^{N-1} \hat{B}_N \end{bmatrix}
$$

is of rank $N^2$. This condition is equivalent to $\mathrm{rank}(\hat{B}_N) = N$, which requires $q \geq N$. Under this condition, to achieve perfect learning as $\hat{m}_N(t) = \hat{m}_N^*(t)$, we can choose the trainable parameters $u(t)$ to satisfy

$$
\hat{B}_N u(t) = \frac{d}{dt} \hat{m}_N^*(t) - \hat{A}_N \hat{m}_N(t). \tag{10}
$$

The condition $q \geq N$ renders this equation to be a underdetermined, and hence has a minimum norm solution, given by,

$$
u^*(t) = \hat{B}_N^T (\hat{B}_N \hat{B}_N^T)^{-1} \Big( \frac{d}{dt} \hat{m}_N^*(t) - \hat{A}_N \hat{m}_N(t) \Big) = \hat{B}_N^T \Big( \frac{d}{dt} \hat{m}_N^*(t) - \hat{A}_N \hat{m}_N(t) \Big).
$$

On the contrary, when $q < N$, then it is impossible for the ECS model to perfectly generate the flow $\hat{m}_N^*(t)$, equivalently, the equation in (10) does not have any solution. Instead, we still have the least-squares solution

$$
u^*(t) = (\hat{B}_N^T \hat{B}_N)^{-1} \hat{B}_N^T \Big( \frac{d}{dt} \hat{m}_N^*(t) - \hat{A}_N \hat{m}_N(t) \Big) = \hat{B}_N^T \Big( \frac{d}{dt} \hat{m}_N^*(t) - \hat{A}_N \hat{m}_N(t) \Big),
$$

giving the minimum value of the loss function. Note that in this case $u^*(t)$ is essentially the first $q$-components of the $N$-dimensional vector $\frac{d}{dt} \hat{m}_N^*(t) - \hat{A}_N \hat{m}_N(t)$. Therefore, with this choices of the training parameters, the moment kernelized ECS model in (9) is reduced to

$$
\frac{d}{dt} \hat{m}_N(t) = \hat{A}_{q,N} \hat{m}_N(t) + \frac{d}{dt} \hat{m}_{q,N}^*(t),
$$

where $\hat{A}_{q,N} \in \mathbb{R}^{N \times N}$ with the first $q$ rows identically 0 and the remaining rows equal to those of $\hat{A}_N$, and $\hat{m}^*_{q,N}(t) = (m^*_0(t), \dots, m^*_q(t), 0, \dots, 0)$. Applying the variation of constants formula yields

$$\hat{m}_N(t) = e^{t\hat{A}_{q,N}} \Big( \hat{m}_N(0) + \int_0^t e^{-s\hat{A}_{q,N}} \frac{d}{dt} m^*_{q,N}(s) ds \Big),$$

so that $\hat{m}^k_N(t) \approx \hat{m}^*_k(t)$ for $0 \leq k \leq q$. Therefore, the training loss satisfies the approximation

$$
\begin{aligned}
\|\hat{m}_N(t) - \hat{m}^*_N(t)\| &\approx \left\| e^{t\hat{\hat{A}}_{q,N}} \big( \hat{m}^*_N(t) - \hat{m}^*_q(t) \big) - \big( \hat{m}^*_N(t) - \hat{m}^*_q(t) \big) \right\| \\
&= \left\| \big( e^{t\hat{\hat{A}}_{q,N}} - I \big) \big( \hat{m}^*_N(t) - \hat{m}^*_q(t) \big) \right\| \\
&\leq \left\| e^{t\hat{\hat{A}}_{q,N}} - I \right\| \left\| \hat{m}^*_N(t) - \hat{m}^*_q(t) \right\| = \Big\| \sum_{k=1}^{\infty} \frac{(t\hat{\hat{A}}_{q,N})^k}{k!} \Big\| \left\| \hat{m}^*_N(t) - \hat{m}^*_q(t) \right\| \\
&\leq \sum_{k=1}^{\infty} \frac{(t\|\hat{\hat{A}}_{q,N}\|)^k}{k!} \left\| \hat{m}^*_N(t) - \hat{m}^*_q(t) \right\| \leq \big( e^{\|\hat{\hat{A}}_{q,N}\|} - 1 \big) \left\| \hat{m}^*_N(t) - \hat{m}^*_q(t) \right\|,
\end{aligned}
$$

where $\hat{\hat{A}}_{q,N}$ is the submatrix of $\hat{A}_N$ consisting of its $q+1$ to $N$ rows and columns. When $N$ and $q$ are large enough, the term $\|\hat{m}^*_N(t) - \hat{m}^*_q(t)\|$ is essentially residue of the Fourier series of the function $x_t - x^*_t$, with $x^*_t$ the density of the desired flow $\mu^*$, which is then approaching to 0.

## C  ADDITIONAL EXPERIMENTAL EVALUATIONS

In this section, additional experimental results are provided to showcase the performance of the proposed ECS model.

### C.1  EXPERIMENTAL DEMONSTRATION OF THE ECS MODEL CAPABILITY

In this section, we use various experiments to demonstrate the excellent model capability of the proposed ECS model. In particular, Figure 3 shows that the ECS model is robust to noise using the CelebA-HQ and AFHA-Cat datasets. In the experiment, we increased the standard deviation of Gaussian noise from 0.1 to 0.4, and the denoised images still have high PSNR values.

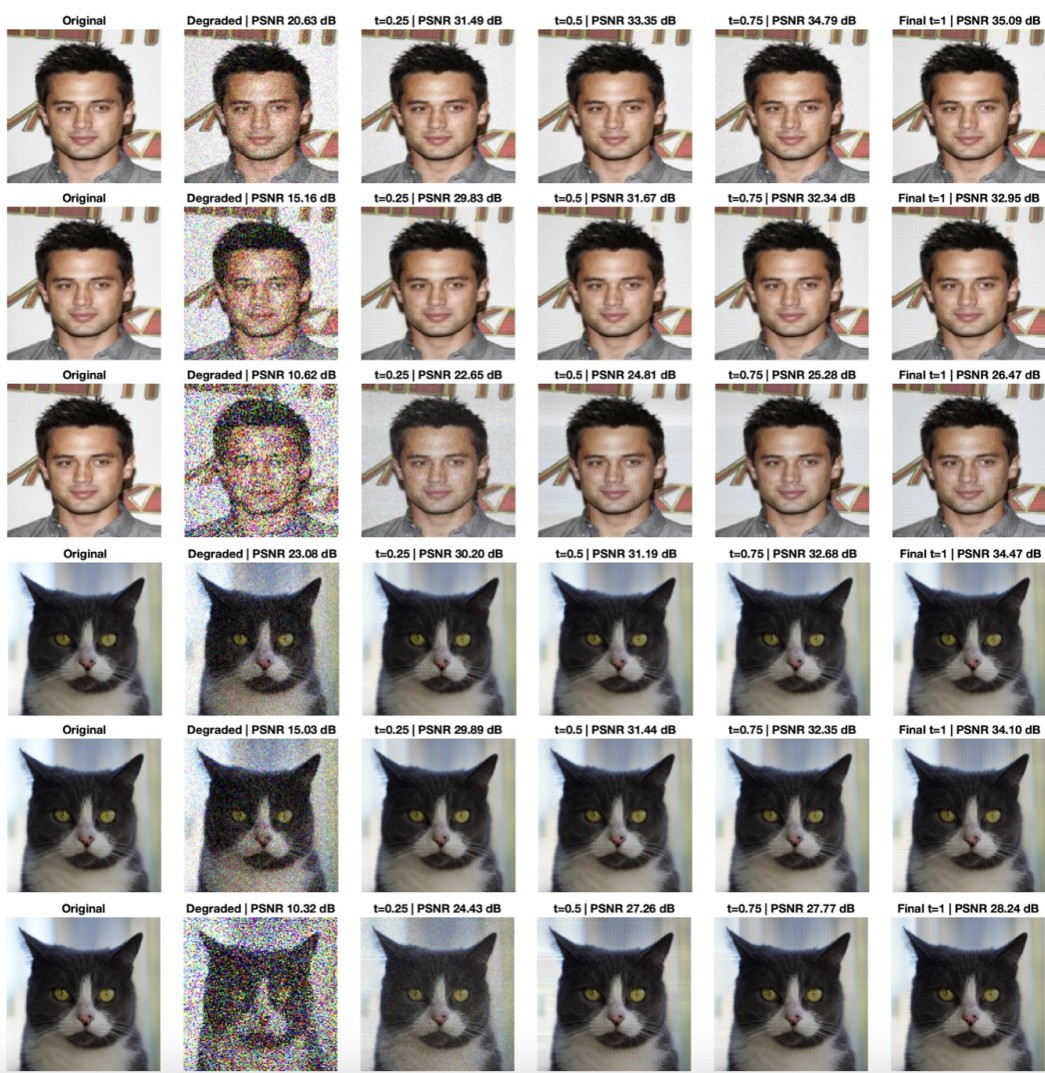

Figure 3: Demonstration of the noise robustness of the ECS model using images in the CelebA-AQ (rows 1 to 3) and ADHQ-Cat datasets (rows 4 to 6). In particular, rows 1 and 4, row 2 and 5, and rows 3 and 6 show the processes of denoising Gaussian noise of standard deviations 0.1, 0.2, and 0.4, respectively. Columns 1 and 2 are the ground truth and noised images; columns 4 to 6 display the output images of the ECS model at time points 0.25, 0.5, 0.75, and 1.

Moreover, the proposed ECS model has the distinguished ability to process high-resolution images. In Figure 4, we show two undistortion processes generated by the ECS model using a $512 \times 512$ image.

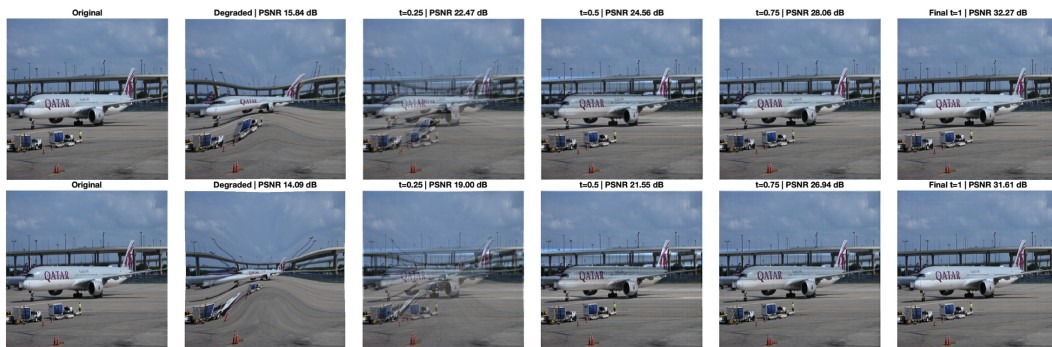

Figure 4: Illustration of the performance of the proposed ECS model on high-resolution images.

## C.2 ADDITIONAL BENCHMARK DATA EXPERIMENTS

We apply the ECS model to two benchmark datasets for image restoration, including AFHQ-cat and LSUN-Bedroom, which consist of RGB images of resolutions $256 \times 256$ and $128 \times 128$, respectively. The simulation results are shown in Figures 5 and 6, respectively. Specifically, in each figure, row 1 illustrates the denoising process; rows 2 and 3 are the deblurring processes for Gaussian blur and motion blur; row 4 is the super-resolution process; and rows 5 to 6 show the processes for inpainting a single box and multiple boxes, respectively.

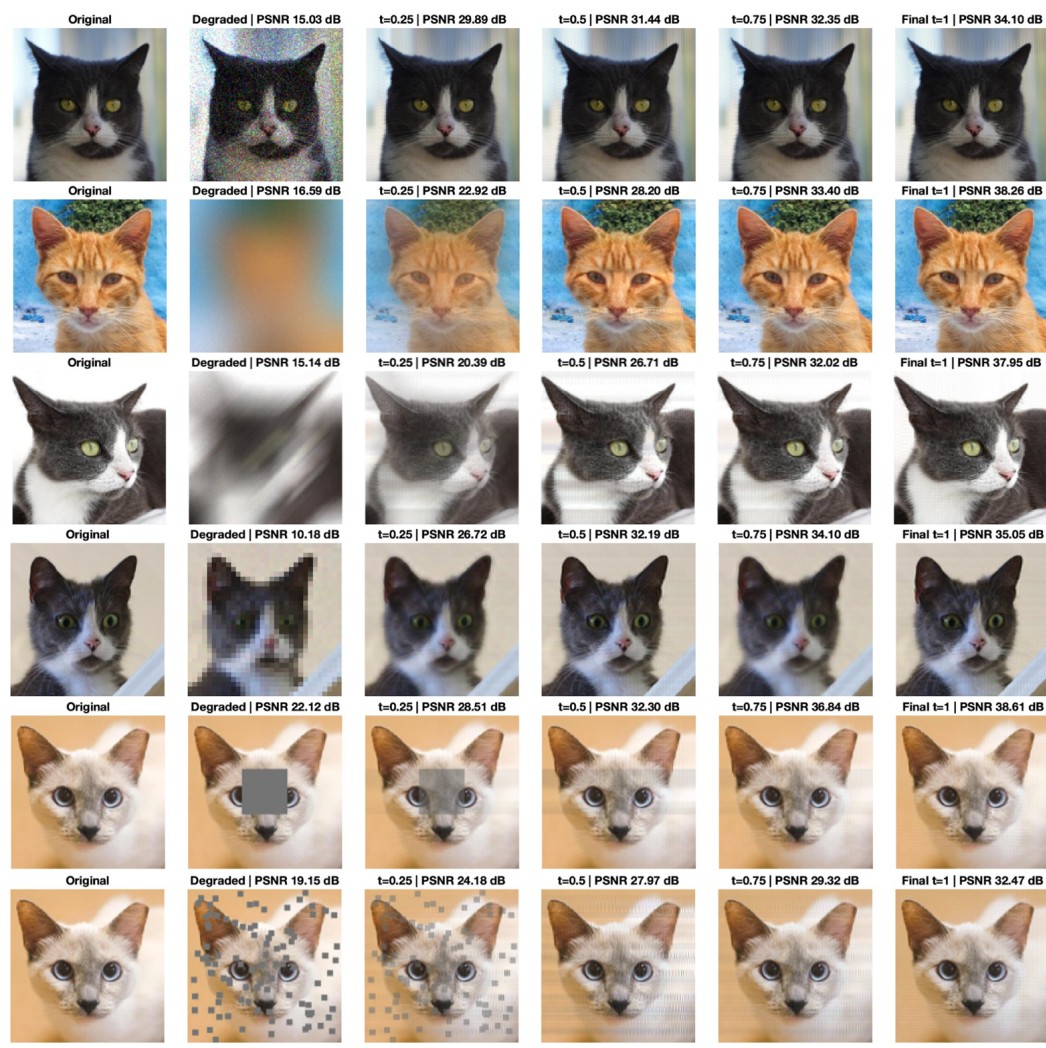

Figure 5: Illustrate of image restoration processes generated by the proposed ECS model for images in the AFHQ-cat dataset. Columns 1 and 2 show the ground truth and degraded images and columns 3 to 6 display the output images of the ECS model at time points 0.25,0.5,0.75 and 1.

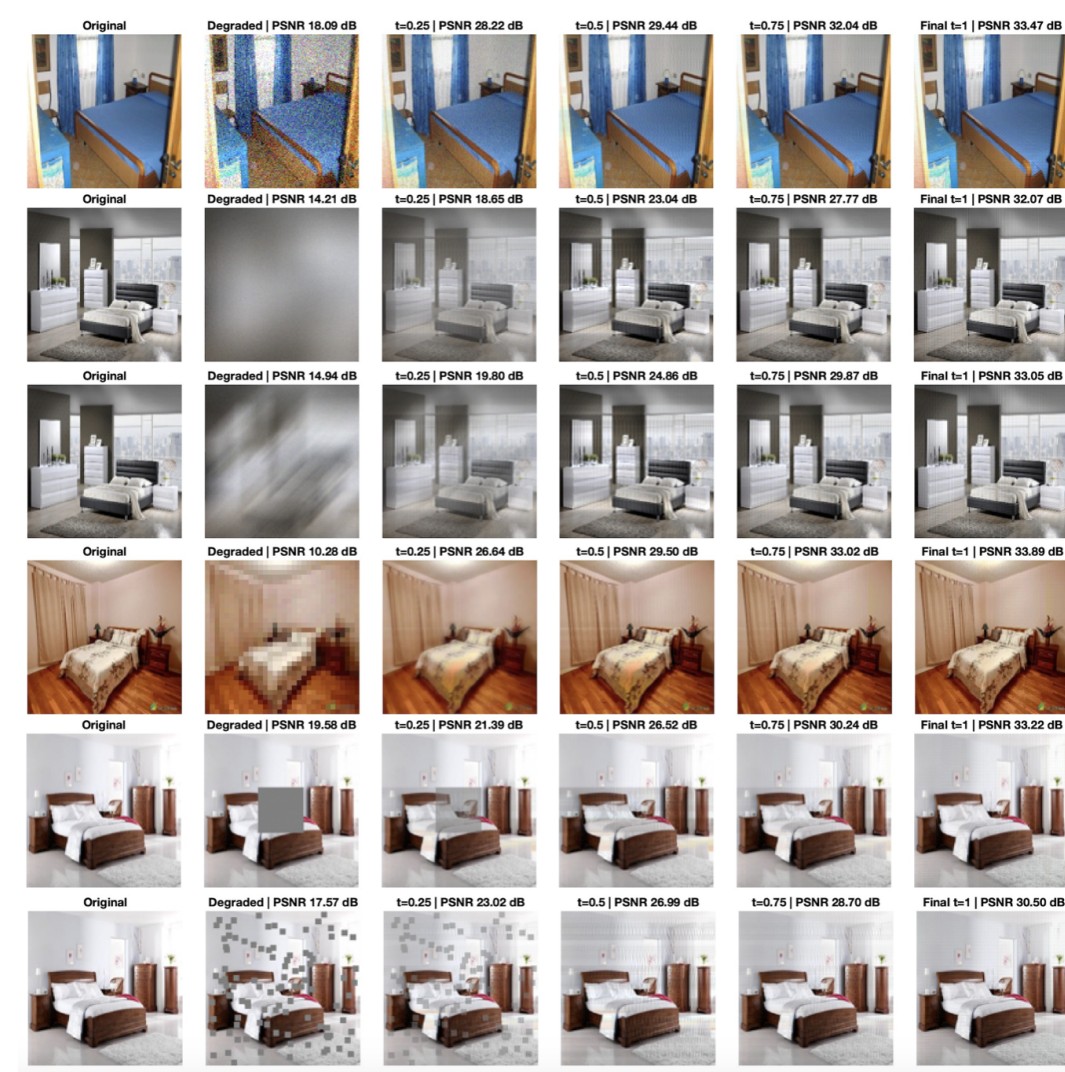

Figure 6: Illustrate of image restoration processes generated by the proposed ECS model for images in the LSUN-Bedroom. Columns 1 and 2 show the ground truth and degraded images and columns 3 to 6 display the output images of the ECS model at time points 0.25, 0.5, 0.75 and 1.

## C.3 ADDITIONAL COMPARISON RESULTS

In this section, we showcase the comparison results between the proposed ECS model and baseline models using the ADHQ-Cat dataset. We consider four image restoration tasks: denoising, deblurring, super-resolution, and inpainting a single mask and multiple masks. For each task, we show the restoration results for a randomly select image, as shown in Figure 7.

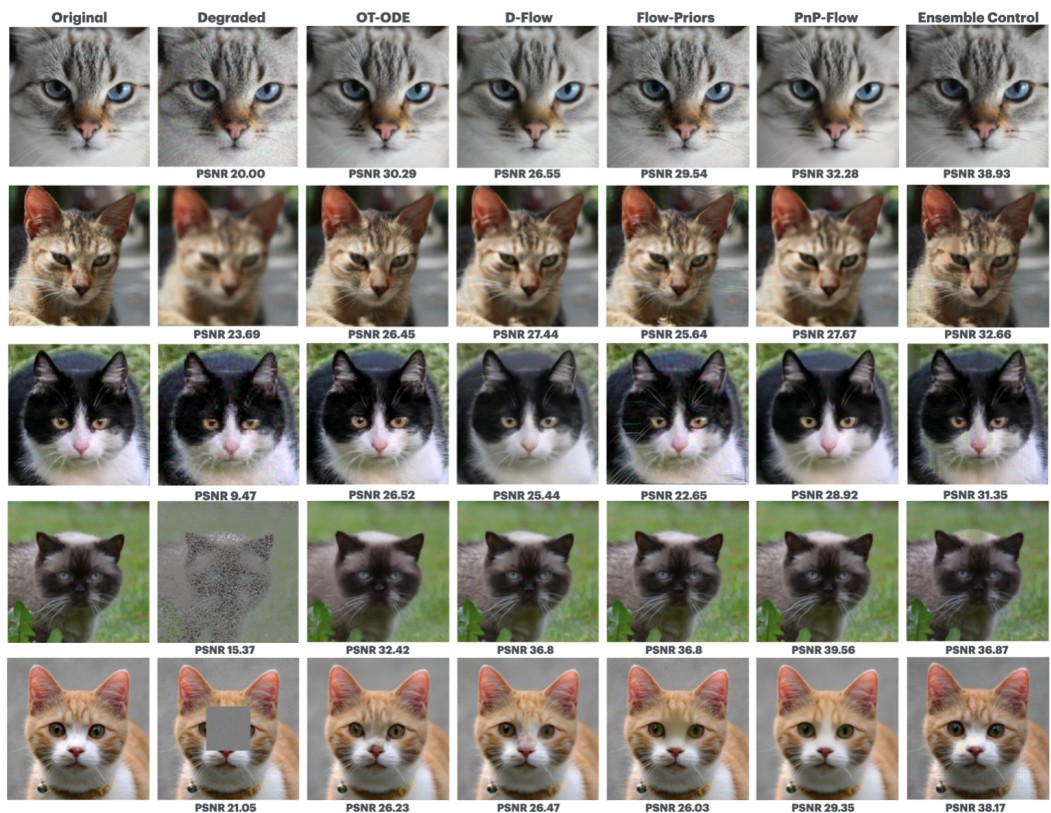

Figure 7: Illustration of the comparison results between the proposed ECS model and baseline models for imaging denoising (row 1), deblurring (row 2), super-resolution (row 3), inpainting of multiple masks (row 4) and a single mask (row 5). Columns 1 and 2 are the ground truth and degraded images; columns 3 to 6 are the restored images output from the baseline models, including OT-ODE, D-Flow, Flow-Priors, and PnP-Flow; column 7 display the images restored by the ECS model.

