# OpenReview forum: "Universal Learning of Distribution Flow using Ensemble Control Systems"
_ICLR.cc/2026/Conference — Submitted to ICLR 2026_

### Official Review · Reviewer_1bwu · 2025-10-26

**Soundness:** 1
**Presentation:** 2
**Contribution:** 2
**Rating:** 2
**Confidence:** 5

**Summary:**

The authors try to develop what they call an ensemble control system (ECS)
by updating the FM dynamics by a system parameter $\beta \in \Omega \subset \mathbb R^d$ and an ''control input'' $u(t) \in \mathbb R^r$:
$$
\frac{d}{dt} x(t,\beta) = v(t,\beta, x(t,\beta),u(t)).
$$
Then they want to introduce a moment kernel representation for learning the model.
Finally, they  compare the performance of their method for different image restoration tasks.

**Strengths:**

The basic idea seems to be interesting, but is hard understand.
The addition of the control $u$ and $\beta$ increases the design space,
which allows for more flexibility.

**Weaknesses:**

The paper is written in an inconsistent way, such that it was impossible for me to judge its correctness.
Neither supplementary material nor a program code was provided which could help here.

The abstract is written in a superlative calling their approach ''exceptionally powerful''.
In view of the bad results with  images having structured artifacts I doubt about this claim.

Introduction: Telling stories about results of Chebyshev and Markov without taking the effort of citing their original paper, but just an overview paper of Mackey is cumbersome.
Concerning generative flows from a dynamic control point of view the authors may have a look at papers of Lorenz Richter et al.

''In general, any pair of distributions with different support cardinalities (reviewer: whatever this means) cannot be transported by FM''
Explain or give a reference.

Section 2:
Let me just start with Subsection 2.1. Here the sample space is first $\mathbb R^d$, but the norm of the sampled $x$ is then in $\mathbb R^n$ and later the authors move to manifolds $M$. They speak about the requirement of finite $k$th moments without specifying the $k$ (reviewer: indeed $k=2$ and the condition is superfluous on compact manifolds). What means ''almost everywhere defined time derivative of a measure $\mu_t$,
in the continuity equation (CE), the $\nabla \cdot$ does not stand for the gradient, as claimed by the authors, but for its adjoint, the divergence.
Further just the (CE) is not enough to make a curve absolutely continuous in the Wasserstein geometry. Further, I am missing a push-forward relation when explaining the connection between the CE and the flow ODE. Then
$$
\frac{d}{dt} x(t,\beta) = v(t,\beta, x(t,\beta))
$$
where $v:[0,1] \times \mathbb R^d \to \mathbb R^n$ makes no sense to me if $d \not = n$.

The section continues in this style.

While the expectation in line 134 is taken with respect to the ''stochastic process'' $x(t)$ it is taken on line 180
''with respect to a probability measure $\lambda$ on $\Omega \subset \mathbb R^d$''. Indeed I cannot interpret minimization problem (4).

Section 3:
Subsection 3.1 starts with that every positive linear functional on $C_c(\Omega)$  corresponds to a measure in $\mathcal P(\Omega)$. However, the dual space of $C_c(\Omega)$ can be identified with all finite Borel measures $\mathcal M(\Omega)$, and a measure is positive if $\langle \mu,\varphi \rangle \ge 0$ for all $\varphi \in C_c(\Omega)$ with $\varphi:\Omega \to \mathbb R_{\ge 0}$. But these are the positive measures, not the probability measures which are in additional bounded. Could the authors explain what they mean?
Then they use $d\mu_t = x_t \, d\lambda$, where $x_t$ is now be the Radon–Nikodym derivative ''of these measures'' and the (reviewer: Schauder) basis of $C_c(\Omega)$ is somehow swiched by  an orhonormal basis of $L^2(\Omega)$ and Plancherel's equality pops up. However, density functions in $L^2(\Omega)$ if $\Omega$ is noncompact are cumbersome.
Then a RKHS is introduced in a non understandable way.

Explain how the dominated convegence theorem is applied in Subsection 3.2.


Section 4:
The final training algorithm is missing. How  problem (7) is solved in practice?
Missing code and incomplete hyperparameters/architectures/training schedules make it difficult to assess correctness or to replicate the reported gains.
Concerning the experiments there is no discussion of the impact of the choice of control $u$;
the images have a lot of artifacts and do not support a significant contribution.
What do the authors attribute the artifacts to? When I look at the inpainting results, I have a small doubt: the image they recover is very similar to the ground truth that I suspect them to have done the control with the test images... (eg. Fig 2 last line, the eyes and nose are super close to ground truth, same for Fig. 7).

**Questions:**

The experiments also do not provide convincing evidence for the correctness and effectiveness of the proposed method. There are so many missing details that it is hard to draw conclusions from the paper:

1. What is the impact of the control $u$?
2. What is the impact of the choice of basis?
3. Why do the images have structured artifacts?
The results are not  convincing. In the evaluation measures the results are not significantly better than related methods, and visually they are worse.
4. What is meant by the statement ''It is worth pointing out that the distribution flows generated by the ECS model are not necessarily
solutions of continuity equations. This indicates that ECS has the ability to learn and generate non-
probability distribution flows. Indeed, when the ensemble state $x_t$ is a general $L^2$-function, rather
than a probability density, the measure $\mu_t$ becomes a signed measure, which can take negative values
on some subsets of $\Omega$. This characteristic allows ECS to handle a broader range of applications
beyond traditional probability distribution flows.''?
5. The run times on page 8 on different machines are not comparable?
6. How does it connect to conditional flow matching?
7. What is the loss you are actually minimizing?
8. The authors compare to FM-based methods that use a generic pre-trained FM model. They should explicit write which pre-trained model they use.
9. ''The number of control inputs [] is chosen to be 150": What does this mean in practice (Network is conditioned on 150 images? If so how those images were chosen? From train or test set?).

---

### Official Review · Reviewer_SGBx · 2025-10-27

**Soundness:** 2
**Presentation:** 2
**Contribution:** 3
**Rating:** 2
**Confidence:** 3

**Summary:**

The paper suggests to use the flow matching velocity fields inside ensemble control systems. Basically, one has a control problem, which is stated as an ODE $\partial_t x(t, \beta) = v(t,\beta, x(t), u(t))$, where one wants to find a control $u$ so that the solution $x$ has desirable properties. This is now cast into a Benamou Brenier style formulation, where we minimize over all controls such that the $\beta$-dependent velocity is close to the flow matching velocity (coming from the cont. equation). This is now turned into a moment equation, and via a parametric ansatz of $x(t)$ in terms of the control u, this can be solved. The approach is tested on image restoration examples and benchmarked against universal restoration method with a pretrained flow matching prior.

**Strengths:**

I believe that the idea is super creative, and could potentially have an enormous impact. For sure, that is not an incremental paper, but rather framing control problems, one can simulate, into minimization problems such that the resulting particle dynamics is close to one, where we now that it has very good properties (the flow matching one). I really like the idea and the way it is framed.

**Weaknesses:**

However, I have several problems with the execution. I first of all have several thing I do not understand, preventing me from giving a high score, since I am not sure how principled the approach is.

1) you write down the minimization problem in (4). For me that is a bit reminiscent of the Benamou Brenier formulation of optimal transport. Is there any theoretical basis why this minimization problem is particularly good to solve to obtain good controls?

2) In 4.1 you make a very specific ansatz for the $x(t)$. In my world, you would now need to prove some kind of universality statement, that your class of $x(t)$ is rich enough to handle interesting control problems. How does this choice of $x(t)$ interact with the abstract given velocity field $v$? You also write that $x(t)$ does not necessarily (or its path) solve the cont equation, but for me this is not a feature of the method.

3) My biggest problem is that you did not apply this to any "interesting" control problems. Moreover, this kind of image generation, is not a control problem. One has a degraded distribution, and you try to get it to the clean image distribution. Now the parameter $\beta$ does not really mean anything. It is a purely auxiliary variable you introduce to fit your framework. This means that you do not really solve a control problem, but just steer between distorted and clean images. I urge the authors to include either a proper control interpretation of this problem or rather test it on more standard (e.g. pde) ones.

4) Furthermore, the baselines are not appropriate. In my opinion, your approach is using "more" information by going from $x_0$ to $x_1$ and a more appropriate baseline would maybe be to take a unet minimizing the L1/L2 loss between degraded and clean. The baselines are all "prior" models, using the flow matching between a gaussian and the data. So their use case is more foundational than what your model is doing.

5) Please discuss the relation to [1], and maybe use it in one of the control experiments, if you do new ones. It is also a control method using flow matching, although much different in spirit, since they guarantee exact boundary condition satisfiability.


[1] Gradient-Free Generation for Hard-Constrained Systems, Cheng et al, ICLR 2025

**Questions:**

1) You write that you use the interpolation $\mu_t = t\ \mu_1+ (1-t)\ \mu_0$, but this is not the flow matching interpolation. Even worse, the resulting velocity would be discontinuous. Are you really using this linear interpolation in the space of measures, or do you rather mean the space of random vectors?

JUSTIFICATION FOR SCORE:

I actually really like the idea, and I am a bit sad I have to give a reject. However, for me both theoretically (why this minimization problem, existence, ansatz) is shaky, and the numerics does not convince me either. If you beef up the numerics, i.e., apply it to more "interesting" control problems maybe with appropriate control baselines, then I will be happy to up my score.

---

### Official Review · Reviewer_6yKo · 2025-10-31

**Soundness:** 1
**Presentation:** 2
**Contribution:** 3
**Rating:** 0
**Confidence:** 3

**Summary:**

This work proposes an alternative way to parameterize the "velocity field" in flow models. Instead of neural nets, this work considers an ensemble control system (ECS) with a kernel representation. Theorem 1 proves the convergence of the kernel representation. Experiments are conducted on repairing noisy, distorted, and damaged images.

**Strengths:**

The methodology would be significant if the presented derivation is true (see weakness). If the presented derivation is true, it provides an alternative way to parameterize a generative flow model without neural nets, which can significantly decrease the deployment cost of some generative models while increasing their accessibility. It's also very interesting and inspiring to see that the RKHS can be "revived" in the deep learning era and work with flow models.

**Weaknesses:**

I don't understand why $ d\mu_t = x_t d\lambda$ around line 219. The Radon–Nikodym density should be a non-negative real-valued function, but $x_t$ takes values from a manifold $M \subset \mathbb{R}^n$  by line 187, so this notation does not make sense to me at all. Even assuming $x_t$ is indeed a Radon–Nikodym density, I think you need a carefully chosen $\lambda$ for $ d\mu_t = x_t d\lambda$, if such $\lambda$ exists. This change of measure seems to be fundamental to the rest of the derivation and proofs. So this question blocks me from appreciating the rest of the paper, and I'm not convinced that the main claim holds.

I'm happy to increase the score if the authors could unblock me.

**Questions:**

See weakness.

---

### Official Review · Reviewer_YQ4T · 2025-11-01

**Soundness:** 2
**Presentation:** 2
**Contribution:** 2
**Rating:** 2
**Confidence:** 3

**Summary:**

This paper proposes a new framework for probability flow learning, i.e.Ensemble Control System (ECS). The authors replace the single unforced dynamic system in Flow Matching (FM) with ECS, transforming the original parameter θ into a control input u(t). To address the infinite-dimensional challenge, they employ a moment kernel transform and achieve efficient approximation through finite-order truncation. Experimentally, ECS is applied to image restoration tasks such as denoising, demonstrating significant improvements over strong baselines including PnP-Flow, Flow-Priors, and D-Flow in metrics like PSNR and SSIM.

**Strengths:**

1. The paper presents an innovative perspective by reformulating the flow matching problem as an optimal control problem from a dynamical systems standpoint.

2. By replacing the single unforced dynamic system in FM with ECS, the method transforms the original parameter θ into a control input u(t). This enhancement significantly improves the model's expressive power, enabling it to handle distributions that FM cannot modeling, such as transitions from Single-center Guassian to multi-center Gaussian distributions.

3. The proposed approach employs moment kernel transform to address the infinite-dimensional challenge, subsequently reducing the problem to a least squares problem.

**Weaknesses:**

1.  I may have completely misunderstood the experimental section, but the purpose of the experimental setup appears unclear to me. Taking the denoising task as an example:
    For the proposed ECS model, the authors seem to treat noisy images as the source distribution and clean images as the target distribution, where each point in these distributions represents a single pixel in the image.  For baseline models, they follow the conventional flow-matching setup, where the source and target distributions consist of noisy/clean images, with each data point being a image.

 This creates an unfair comparison:
   The ECS model more like a supervised method by operating at the pixel level with paired supervision.  The baselines are unsupervised methods having to learn the mapping at the image level without the same supervision.
   These two paradigms are fundamentally incomparable.

2.  I also fail to understand how the proposed algorithm could be applied to real-world denoising tasks.  In practice, we do not have access to the ground-truth clean images corresponding to noisy input. This is precisely the denoising aims to solve.  It's unclear how the method would perform in image denoising by the proposed method.

**Questions:**

The paper needs a fair comparison with the other method.

The paper assumes paired distributions (noisy → clean), but this supervision is unavailable in real scenarios.

---

### Meta-Review · Area_Chair_geLQ · 2026-01-05

**Summary:**

The paper introduces a control-based approach to generative flow matching using an Ensemble Control System (ECS). While reviewers acknowledged its innovative potential to address limitations in existing flow-based models, they expressed significant concerns regarding theoretical rigor, the validity of the theoretical claims and formulation of the approach, and insufficient experimental validation. These issues remained unresolved as the authors did not submit a rebuttal. Therefore, I recommend rejecting the paper.

**Reviewer Concerns:**

The reviewers raised concerns about several aspects of the paper, including readability, theoretical soundness, experimental design, and validation of the proposed approach. Specifically, reviewers 6yKo, SGBx, and 1bwu highlighted issues with the theoretical formulation, inconsistencies, and potential technical flaws. Reviewer YQ4T questioned the formulation of the denoising problem, while reviewers SGBx and 1bwu also expressed concerns regarding the overall experimental design and validation and poor comparison with other relevant approaches. None of these concerns were addressed since the authors didn't provide any rebuttal.

**Reviewer Scores:**

The authors didn't provide any  rebuttal and  no discussion took place.

---

### Decision · Program_Chairs · 2026-01-26

Reject